# Reassessment of Reliability and Reproducibility for Triple-Negative Breast Cancer Subtyping

**DOI:** 10.3390/cancers14112571

**Published:** 2022-05-24

**Authors:** Xinjian Yu, Yongjing Liu, Ming Chen

**Affiliations:** 1Department of Bioinformatics, College of Life Sciences, Zhejiang University, Hangzhou 310058, China; xyu16@zju.edu.cn (X.Y.); coy@zju.edu.cn (Y.L.); 2Quantitative and Computational Biosciences Graduate Program, Baylor College of Medicine, Houston, TX 77030, USA; 3Bioinformatics Center, The First Affiliated Hospital, School of Medicine, Zhejiang University, Hangzhou 310058, China

**Keywords:** triple-negative breast cancer, molecular subtype, subtyping benchmark, microarrays, clustering, biomarker discovery, pipeline

## Abstract

**Simple Summary:**

Triple-negative breast cancer (TNBC) is a heterogeneous disease. A proper classification system is needed to develop targetable biomarkers and guide personalized treatment in clinical practice. However, there has been no consensus on the molecular subtypes of TNBC, probably due to discrepancies in technical and computational methods chosen by different research groups. In this paper, we reassessed each major step for TNBC subtyping and provided suggestions, which promote rational workflow design and ensure reliable and reproducible results for future studies. We presented a recommended pipeline to the existing data, validated established TNBC subtypes with a larger sample size, and revealed two intermediate subtypes with prognostic significance. This work provides perspectives on issues and limitations regarding TNBC subtyping, indicating promising directions for developing targeted therapy based on the molecular characteristics of each TNBC subtype.

**Abstract:**

Triple-negative breast cancer (TNBC) is a heterogeneous disease with diverse, often poor prognoses and treatment responses. In order to identify targetable biomarkers and guide personalized care, scientists have developed multiple molecular classification systems for TNBC based on transcriptomic profiling. However, there is no consensus on the molecular subtypes of TNBC, likely due to discrepancies in technical and computational methods used by different research groups. Here, we reassessed the major steps for TNBC subtyping, validated the reproducibility of established TNBC subtypes, and identified two more subtypes with a larger sample size. By comparing results from different workflows, we demonstrated the limitations of formalin-fixed, paraffin-embedded samples, as well as batch effect removal across microarray platforms. We also refined the usage of computational tools for TNBC subtyping. Furthermore, we integrated high-quality multi-institutional TNBC datasets (discovery set: *n* = 457; validation set: *n* = 165). Performing unsupervised clustering on the discovery and validation sets independently, we validated four previously discovered subtypes: luminal androgen receptor, mesenchymal, immunomodulatory, and basal-like immunosuppressed. Additionally, we identified two potential intermediate states of TNBC tumors based on their resemblance with more than one well-characterized subtype. In summary, we addressed the issues and limitations of previous TNBC subtyping through comprehensive analyses. Our results promote the rational design of future subtyping studies and provide new insights into TNBC patient stratification.

## 1. Introduction

Triple-negative breast cancer (TNBC) accounts for 10–20% of primary breast cancers that lack expression of estrogen receptor (ER), progesterone receptor (PR), and human epidermal growth factor receptor 2 (HER2) [1]. TNBC is highly invasive and metastatic, prone to relapse, and has a poor prognosis. Moreover, due to its molecular phenotype, TNBC is not sensitive to anti-estrogen/androgen hormone therapies or HER2-targeted therapy [2]. Since it is a heterogeneous group without specific molecular features, TNBC varies in prognosis and response to treatment [3]. Therefore, a classification system for TNBC is urgently needed to develop potential biomarkers or drug targets and to promote personalized care for TNBC patients.

Several studies have identified molecular subtypes of TNBC based on gene expression profiles. Lehmann et al. [4] used publicly available expression data to divide TNBC into six subtypes: basal-like 1 (BL1), basal-like 2 (BL2), mesenchymal (M), mesenchymal stem-like (MSL), immunomodulatory (IM), and luminal androgen receptor (LAR). Later, they refined the classification into four subtypes (BL1, BL2, M, and LAR) by histopathological quantification and laser-capture microdissection [5]. Burstein et al. [6] first used immunohistochemistry (IHC)-confirmed TNBC samples to get four subtypes with distinct prognosis: LAR, M, basal-like immunosuppressed (BLIS), and basal-like immune-activated (BLIA). Liu et al. [7] considered long noncoding RNAs (lncRNAs) and got a similar four-subtype classification. Jézéquel et al. [8,9] used fuzzy clustering and identified three subtypes: LAR and two basal-like subtypes with different immune responses. Other studies explored TNBC classification based on immune signatures [10,11] and microenvironment features [12]. Although several studies have investigated TNBC subtypes, their sample quality and chosen methods vary. The lack of consistency in TNBC subtyping methods and results is a huge barrier to developing standards and guidelines for researchers and clinicians.

The primary purpose of this work is to provide perspectives on the molecular subtyping of TNBC and to help with the rational design of studies. We present a modularized pipeline for the analysis and reassess widely used methods in each step (Figure 1). The pipeline is applied on selected publicly available datasets to validate TNBC subtyping results with a larger sample size.

## 2. Materials and Methods

### 2.1. Data Collection

We collected 11 publicly available breast cancer datasets (Appendix A) from the Gene Expression Omnibus database (GEO, https://www.ncbi.nlm.nih.gov/geo/, accessed on 20 December 2020). Seven of them were IHC-confirmed TNBC microarray datasets, in which we evaluated the effects of sample preservation and batch effect removal methods on TNBC subtyping. Our criteria for selecting IHC-confirmed TNBC datasets were as follows: (1) sufficient sample size; (2) explicit statement of the sample preservation methods; (3) preferably with previous TNBC subtyping results so that we could assess the reproducibility. Our own TNBC subtyping was based on the four datasets with fresh-frozen (FF) tissues: GSE103091, GSE103668, GSE76124 (discovery set), and GSE76250 (validation set).

### 2.2. Microarray Data Preprocessing

Microarray data was RMA normalized and log transformed using the R package *oligo.* For the Affymetrix Human Transcriptome Array (HTA) 2.0 platform [13], control probes were removed using R package *affycoretools*. For one probe targeting multiple genes, the first gene in the annotation was chosen. For genes with multiple probes, the probe with the highest average expression across samples was chosen to represent the gene. Probe ID was converted to Entrez Gene ID and Gene Symbol. The batch effect was removed using the *removeBatchEffect* function in the R package *limma* [14].

### 2.3. Gene Selection and Subtype Discovery

The most differentially expressed genes with the top 10% standard deviations were selected for the discovery set and validation set, respectively. Then, we found their union set for clustering to ensure that the same genes were used as clustering features. We performed consensus clustering using *k*-means with 1000 iterations independently on the discovery set and validation set. The optimal number of clusters (*k*) was determined by the cumulative distribution function (CDF) curve, where the increase in *k* does not result in a substantial increase in the relative area change under the CDF curve. Heatmaps were generated using the R package *ComplexHeatmap* [15] to visualize the results.

### 2.4. Differential Expression and Pathway Enrichment Analysis

We used the R package *limma* [13] to find differentially expressed genes (DEGs). After correction of false discovery rate (adjusted *p* value < 0.05), genes with fold change ≥1.5 were considered to be significantly differentially expressed. For each subtype, we compared it with the rest samples to identify DEGs specific to this certain subtype. GO pathway enrichment analysis was performed using Metascape 3.5 (metascape.org, accessed on 15 April 2021) [16].

### 2.5. Comparison with Established Subtypes

The R package *genefu* [17] was used to assign samples to intrinsic subtypes of breast cancer. We chose the PAM50 [18] classifier with the robust scaling method in our analysis. We obtained Lehmann’s classification using the TNBCtype tool [19] and labeled those samples that could not pass the ER-positive filter as an additional group. Burstein et al. [6] provided an 80-gene centroid signature of the 4 subtypes they discovered, which we downloaded from their Appendix A. Using this signature, we have assigned all samples to Burstein’s subtypes. Specifically, we used the expression values of the 80 signature genes in our datasets and calculated a Pearson correlation coefficient with the centroid values. For each sample, a Burstein’s subtype was assigned based on the highest Pearson correlation coefficient found among the 4 subtypes.

### 2.6. Survival Analysis

We used the dataset GSE103091, which has available metastasis-free survival (MFS) records, to compare prognosis differences among the discovered subtypes. Survival curves were established by the Kaplan–Meier method. The log-rank test was used to test survival differences between the subtypes. Survival analysis was performed using the R package *survival*.

### 2.7. Microenvironment Cell Abundance Calculation

We used the tool CIBERSORT [20] to estimate the abundance of 22 types of tumor-infiltrating immune cells with its default signature gene file LM22 and permutation number of 100. Since LM22 has not been validated on the microarray platform HTA 2.0 and may produce unreliable results, we eliminated the dataset GSE76250 from this analysis.

## 3. Results

### 3.1. Intrinsic Subtype Distributions Are Impacted by Sample Preservation Methods

RNA profiling requires nucleic acid extraction from FF or formalin-fixed, paraffin-embedded (FFPE) tissues. Although FFPE samples make long-term storage and retrospective analysis possible, the quality of nucleic acids in FFPE samples is far from optimal due to chemical cross-linking and nucleic acid fragmentation [21]. Therefore, we evaluated whether sample preservation methods could affect TNBC subtyping results.

When performing intrinsic subtyping [18] on the seven publicly available IHC-confirmed TNBC datasets (Appendix A), we found that datasets using FFPE tissues had low basal-like proportions (Figure 2A), which is inconsistent with the consensus that ~75% TNBCs are basal-like [22]. Surprisingly, GSE76250 has a much lower proportion (~50%) of basal-like tumors than the original results (60–80%) [12], possibly because almost one-third of samples were excluded from their analysis. There was not enough information to explain how and why they excluded these samples, so we retained all of the data in our analysis.

To further investigate whether sample preservation methods influence gene expression and affect basal-like proportions, we performed differential gene analysis between the basal-like tumors in FFPE and FF samples. Four more public breast cancer datasets were included (Appendix A). Differentially expressed genes (adjusted *p* value < 0.05 and fold change ≥ 1.5) were identified independently for each microarray platform (Figure 2B). Using the Affymetrix Human Genome U133 Plus 2.0 Array (U133 Plus 2) platform, FFPE samples had 17 upregulated and 33 downregulated genes; for the Affymetrix Human Transcriptome Array 2.0 (HTA 2.0) platform, FFPE samples had 27 upregulated and 11 downregulated genes. In addition, 6 out of 88 differentially expressed genes (ESR1, FOXA1, SFRP1, MELK, CDC20, and NUF2) overlapped with the PAM50 gene set. The results indicated that sample preservation methods may influence subtype distributions by changing the expression patterns of some essential genes. Therefore, we suggest that researchers pay attention to this issue when using FFPE samples.

### 3.2. Cross-Platform Batch Effect Removal Results in Less Stable Clusters

When datasets from multiple centers, cohorts, or array platforms are used in a single analysis, it is necessary to consider batch effects in data preprocessing [23]. Principal component analysis (PCA) showed that each dataset we used formed a separate cluster without batch effect removal (Figure 2C). When these datasets are used for TNBC subtype discovery, batch effects may lead to problematic group assignment that relies largely on technical differences among datasets instead of meaningful biological differences. Therefore, adjustment for batch effects before clustering is generally required when such effects exist.

To address this issue, several tools have been developed and are in common use, such as the *removeBatchEffect* function (using a two-way ANOVA) in the *limma* package [14] and the *Combat* function (using an empirical Bayes method) in the *sva* package [24]. However, evidence shows that these methods can also exaggerate group differences when the group–batch distribution is unbalanced [25]. It is important to note that batch effect removal requires additional consideration in particular cases.

Since data distribution differs greatly among platforms (Figure 2D), we evaluated whether batch effect removal across different microarray platforms would affect downstream clustering analysis for TNBC subtyping. For comparison, we conducted batch effect removal in two different ways: (1) concatenating datasets from different platforms, removing batch effect, and performing clustering as a single dataset; or (2) concatenating datasets with the same platform, removing batch effect independently for each platform, and performing clustering separately for each platform. When compared to combining datasets from different platforms, removing batch effects separately for each platform showed a significantly higher average silhouette width (Wilcoxon signed-rank test: *p* = 2.2 × 10^−16^), indicating more stable clustering results (Figure 2E). Therefore, we suggest removing batch effects separately for different microarray platforms in TNBC subtyping.

### 3.3. Proper Clustering Features and Algorithms Should Be Chosen for Subtype Discovery

Choosing proper features for unsupervised clustering is an important step in TNBC subtype discovery. Gene expression is the most commonly used feature. Generally, genes with standard deviations exceeding a certain threshold are selected for clustering [4,7,8]. Features derived from raw gene expression profiles can also be used. For instance, pathway-level features quantified by algorithms such as single-sample gene-set enrichment analysis (ssGSEA) could produce more stable and reproducible cancer classification results [26,27]. Besides, certain features can be used to investigate specific questions, e.g., immune-related pathway signatures [10,11] or immune infiltration cell abundance [12] help to identify TNBC subtypes that may be responsive to immunotherapy. Moreover, gene expression profiles can be combined with other information, such as the protein interaction network, to develop new features and to obtain more stable and comprehensive clustering results [28].

Several clustering algorithms have been used for TNBC subtype discovery (Table 1). The performance of common clustering algorithms has been evaluated on cancer subtyping problems. A large-scale analysis of 7 clustering methods and 35 cancer microarray datasets revealed that the finite mixture of Gaussians and *k*-means showed the best clustering results [29]. The hierarchical methods, although widely used in biomedical literature, performed poorly when compared to other methods [29]. Brunet et al. [30] successfully applied non-negative matrix factorization (NMF) on cancer microarray data to identify molecular patterns and demonstrated its advantages over hierarchical clustering and self-organizing maps (SOMs). However, NMF requires intensive computations, which are highly time-consuming for large-scale expression data [31].

Besides the above-mentioned algorithms, consensus clustering is an analogy to ensemble learning methods in supervised learning. The clustering result is the consensus over multiple runs of a basic algorithm, such as *k*-means and hierarchical clustering [38]. Consensus clustering is more robust and shows advantages for discovering biological meaningful clusters [38]. To improve the stability of the clusters, we suggest considering consensus clustering instead of using results directly from a single run.

### 3.4. Existing Algorithms Could Be More Suitable for TNBC Subtype Prediction than the Simple Nearest Centroid Classifier

After identifying subtypes in the discovery set, it is necessary to reproduce the classification on the validation set or to develop a subtyping tool for unclassified datasets. Prediction algorithms to label independent datasets with the discovered subtypes are presented in Table 2. The most straightforward method is the simple nearest centroid classifier (SNCC). It computes centroids for each class in the discovery dataset and assigns a new sample to one of the classes based on the lowest Euclidean distance or the highest correlation coefficient. This method is used in the web-based tool TNBCtype for Lehmann’s classification of TNBC [19] and the R package *genefu* for intrinsic subtyping of breast cancers [17].

There are modified centroid-based methods that further improve SNCC. Prediction Analysis of Microarrays (PAM) [39] shrinks the centroids and gives higher weights to genes that are more stable within samples of the same class. Compared with SNCC, PAM demonstrates higher prediction accuracy with only a small subset of centroids. A study on breast cancer classification also confirmed PAM’s superiority over SNCC [41]. Classification to Nearest Centroids (ClaNC) [40] aims to be simpler and even more interpretable than PAM. It ranks genes by standard *t*-statistics but does not shrink centroids. It is shown that the prediction error rate of ClaNC is significantly lower than that of PAM.

In addition to centroid-based methods, supervised machine learning could also be applied. Several machine learning methods on multiclass cancer subtyping have been assessed [42]. It is indicated that the support vector machine (SVM) is the best classifier. *k*-nearest neighbor (*k*-NN) also achieves satisfactory accuracy on most datasets. Both of them outperformed decision trees [42]. Simple classifiers, such as linear discriminant analysis (LDA) and *k*-NN, performed markedly better than more sophisticated algorithms such as aggregated classification trees [43].

Although widely used in existing tools, SNCC gives each feature the same weight, which may not be suitable for cancer subtyping. Specific genes in the centroids may contribute more to distinguishing different subtypes [44]. To obtain better prediction results, further research could focus on testing modified centroid-based methods or machine learning algorithms such as SVM, LDA, or *k*-NN when developing TNBC subtyping tools.

### 3.5. Intrinsic Subtypes Could Not Be Reliably Assigned Due to Large Discrepancy of Different Gene Sets

In 2000, Perou et al. [45] first identified five intrinsic molecular subtypes for breast cancer (luminal A, luminal B, HER2-enriched, basal-like, and normal-like) by gene expression profiling. The ‘intrinsic’ gene set used to discover these subtypes is defined as genes for which expression varies significantly greater between tumor samples from different patients than between paired samples from the same person. Later, the intrinsic subtyping system was validated on independent datasets, which confirmed its clinical significance [46]. The intrinsic gene set was modified several times by Sorlie et al. in 2003 [47] and Hu et al. in 2006 [48]. In 2009, Parker et al. [18] developed a 50-gene classifier called PAM50, which further reduced the intrinsic gene number to 50 and made it more efficient for clinical testing. Since the intrinsic classification is well established and has a clinical impact, many studies on TNBC subtyping compare it with their results. A useful tool for intrinsic subtyping is the R package *genefu* [17].

In *genefu*, the intrinsic subtyping function requires two main parameters: the centroid gene set and the gene expression standardization method (without scaling, traditional scaling, or robust scaling). The traditional scaling method is based on mean and standard deviation, while the robust scaling method is based on the 0.025 and 0.975 quantiles. The robust method has been tested in numerous breast cancer samples and reached the best concordance with the clinical test [17]. To provide suggestions for the proper use of *genefu*, we evaluated the effect of the two parameters on the subtyping results. The subtyping results without scaling were distinctly unstable when compared with standard or robust scaling. Therefore, performing intrinsic subtyping without scaling the gene expression values is not recommended (Figure 3A).

We then evaluated the effect of the two parameters independently. When fixing the gene set, the majority of the samples (over 80%) were assigned to the same subtype with a different scaling method. When fixing the scaling method, the results of the three gene sets were identical in more than 50% of the samples, but ~20% of the samples were assigned to completely different subtypes (Figure 3B). The choice of gene sets produced significantly different subtype distributions (Chi-squared test: *p* = 1.2 × 10^−14^, 1.5 × 10^−10^ for using standard and robust scaling, respectively). It shows that changing gene sets has a greater impact on the subtyping results than using different scaling methods, probably because of the large discrepancy among the three gene sets (Figure 3C). Our results indicated that different gene sets do not reliably assign the sample patients to the same intrinsic subtypes, which is consistent with a previous study [49]. Therefore, the use of microarray gene profiling needs more stringent standardization of methodologies.

Interestingly, although the genes have few overlaps among the three gene sets, they are enriched in almost identical biological pathways (Figure 3D), probably due to the fact that different intrinsic gene sets focused on similar biological processes. We suggest developing pathway-based intrinsic subtyping methods, which may produce more robust results than gene-based methods.

Notably, Parker et al. [18] identified four additional groups that could possibly represent intermediate states or ambiguity in their clustering. We also noticed that some samples had very similar correlation coefficients for several subtypes, and so the highest correlation may not be a reliable assignment indicator. Nevertheless, the function in *genefu* does not label any sample as ‘unclassified.’ We recommend considering the possible ambiguity of the intrinsic classification and further process the subtyping results to find those unclassifiable samples.

### 3.6. mRNA-Based ER-Positive Filters Cause Unwanted Exclusion of IHC-Confirmed TNBC Samples

Lehmann et al. [4] first identified the molecular subtypes of TNBC and developed a web-based tool TNBCtype to perform their classification, which is now widely used in subtyping studies of TNBC [19]. However, several key issues may affect the usage of the tool.

First, Lehmann’s classification was modified in 2016, when two of the original six subtypes were no longer retained [5]. However, the tool was not updated accordingly and thus later studies were not aware of the change and still used six subtypes [9,12]. Therefore, we recommend reanalyzing the results of TNBCtype to get the updated four-subtype classification. Our re-analyzation code is available in the Appendix A for reference.

Second, before performing the analysis, TNBCtype filters the expression matrix by checking if there are ER-positive samples (defined as samples that have ER expression values greater than the upper quartile of the gene expression values). However, when we uploaded the datasets with only IHC-confirmed TNBC samples (GSE76124 and GSE76250), a large proportion of samples still did not pass this quality assessing step. The problem lies in Lehmann’s filtering method. Instead of IHC detection, which is the “gold standard” in clinical situations [4], TNBCtype applied an in-house filter based on mRNA expression. Although TNBC patients do not express ERα, ~30% of them overexpress another form of the estrogen receptor, ERβ [50]. ERβ can form functional heterodimers with androgen receptor (AR), which might produce TNBC samples with ER-positive-like gene expression patterns [51]. These mechanisms suggest that the ER filter of TNBCtype is problematic and may skew the subtyping results.

### 3.7. Unsupervised Clustering Reveals Six Molecular Subtypes of TNBC

We selected four public TNBC datasets with FF tissues and divided them into discovery (GSE103091, GSE103668, and GSE76124) and validation sets (GSE76250) according to different microarray platforms. We then performed consensus clustering independently on the two sets. Using *k*-means clustering with 1000 iterations, the relative area changes under the consensus distribution function (CDF) curve indicated the optimal number of clusters was 6 [38] (Figure 4A).

Differentially expressed genes (adjusted *p* value < 0.05 and fold change ≥ 1.5) for each subtype were identified and visualized (Figure 4B). The six subtypes showed consistent expression patterns between the discovery set and validation set, indicating reproducible clusters. Pathway enrichment analysis on these differentially expressed genes confirmed the molecular characteristics of each subtype. In conclusion, we validated four previously defined subtypes (Subtypes 1, 3, 4, and 5) and discovered two more subtypes (Subtypes 2 and 6).

Subtype 1 is characterized by the downregulation of immune-related pathways, such as leukocyte migration, inflammatory response, and interleukin production. Meanwhile, this subtype shows the upregulation of cell cycle-regulated pathways, indicating its highly proliferative nature. Intrinsic subtyping with PAM50 showed that Subtype 1 tumors are mostly basal-like (Figure 4C). TNBC subtyping with Burstein’s centroids confirmed that Subtype 1 corresponds to the BLIS subtype of previous studies [6,7] (Figure 4C).

In contrast to the BLIS subtype, Subtype 3 exhibits upregulated immunoregulation pathways, including the activation of B cell, T cell, and natural killer cell-regulated pathways. Thus, we consider Subtype 3 as the IM subtype [4,7]. Subtyping with Burstein’s centroids and TNBCtype showed that the previously discovered BLIA subtype [6] is basically the same as the IM subtype (Figure 4C).

Subtype 4 tumors show enrichment in steroid hormone-related pathways. Although confirmed by IHC as ER-negative, this subtype has the upregulation of AR, ESR1, and other estrogen-regulated genes such as FOXA, XBP1, and GATA3. These features suggest that subtype 1 is the previously defined LAR subtype [4,6] (Figure 4C). The LAR subtype may respond to anti-estrogen or anti-androgen therapies. Since this subtype has ER-positive-like expression patterns, many samples that are classified as the LAR subtype could not pass the ER-positive filter when using TNBCtype (Figure 4C). The results suggested that this filtering step is not necessary and may skew the subtype distribution.

Subtype 5 shows a variety of upregulated pathways, including extracellular structure organization, response to growth factor, and lipid metabolic process. In addition, cell cycle-related pathways are inhibited in subtype 5. This subtype matches the description of the MES or MSL subtype in previous studies [4,6,7], which can also be supported by subtyping with Burstein’s centroids and TNBCtype (Figure 4C). We finally labeled it as the MES subtype in the subsequent analysis. Interestingly, many MES tumors also could not pass the ER-positive filter of TNBCtype. Further research could focus on finding out why some of the MES tumors produce ER-positive-like expression patterns.

In addition to these previously defined subtypes, we also identified two clusters (Subtype 2 and 6) that have not been characterized in previous studies, which possibly represent intermediate states of tumors considering their resemblance of expression patterns with the other four subtypes. Therefore, we termed them as the INT1 and INT2 subtype, respectively. The INT1 subtype has similar expression patterns as the BLIS subtype, except that the INT1 subtype does not show immune response inhibition. The INT1 subtype is characterized by the upregulation of MUC16, TMPRSS3, and ART3. MUC16 is a well-known cancer biomarker and has becomes a potential target for therapy in recent years [52]. The INT2 subtype shares characteristics with both the BLIS (upregulation of cell cycle-related pathways) and MES (upregulation of extracellular matrix-related pathways) subtypes. The INT2 subtype is characterized by the upregulation of CXCL13, CXCL9, and POU2AF1. The upregulation of immune-related genes indicates a certain extent of immune activation in INT2.

The INT1 and INT2 subtypes might be easily mixed up with the other four well-defined subtypes since they do not have distinct expression features. However, they differ in some important characteristics and should not be simply assigned to the other four subtypes. These features, such as immune response and the upregulation of specific genes, have potential clinical implications that could help guide proper treatment for TNBC patients with an ambiguous classification.

### 3.8. TNBC Subtypes Stratify Patients’ Survival

Clinical outcome data were available for dataset GSE103091. Analysis of MFS showed a trend in survival difference among six subtypes (Figure 5A), although it did not reach significance (log-rank test: *p* = 0.2). The IM subtype and the BLIS subtype tended to have the best and the worst prognosis, respectively, which could be explained by their molecular characteristics. Immune suppression and upregulated proliferation-related pathways are possibly the cause of the poor prognosis of the BLIS subtype. On the contrary, the activation of various immune-related pathways in the IM subtype leads to a good prognosis. Previous studies also found similar survival differences for the IM [6] and the BLIS subtypes [6,7].

Interestingly, the INT1 and INT2 subtypes also tended to have better survival than the BLIS subtype (Figure 5A), although they sharing some features in their expression patterns. It is an indicator that differentiating these two subtypes from the other four stable subtypes could have clinical implications.

It should be noted that the non-significant statistical result is probably due to the lack of available follow-up clinical outcome data and the uneven distribution of samples among the six subtypes. However, this trend for the prognostic difference seems to be meaningful and explainable. Future research could test this in larger cohorts.

### 3.9. TNBC Subtypes Differ in Microenvironment Phenotypes

Since intratumor response plays a vital role in basal-like tumors and has prognostic significance, we focused on the immune response of the four basal-like subtypes: BLIS, IM, INT1, and INT2. We calculated the composition of infiltrating immune cells by CIBERSORT. After excluding immune cell types with a median proportion value lower than 0.01, eight were kept for the analysis and were assigned to pro-tumorigenic (associated with better survival in TNBC patients) and anti-tumorigenic (associated with worse survival in TNBC patients) based on previous studies [9]. In concordance with previous studies, BLIS tumors were highly infiltrated by the three pro-tumorigenic cell types (Figure 5B) and were associated with worse survival (Figure 5A). On the contrary, IM tumors were highly infiltrated by four anti-tumorigenic cell types and were associated with improved survival. INT1 tumors, which were mostly Burstein’s BLIA subtype (Figure 4C), had a significantly higher abundance of T follicular helper cells (Tfhs) (Wilcoxon signed-rank test: *p* = 2.3 × 10^−9^ and a lower abundance of resting mast cells (Wilcoxon signed-rank test: *p* = 1.4 × 10^−6^) as compared to IM tumors. Similarly, INT2 tumors that were mostly Burstein’s BLIS subtype, had a significantly lower abundance of Tfhs (Wilcoxon signed-rank test: *p* = 2.5 × 10^−4^) and a higher abundance of resting mast cells (Wilcoxon signed-rank test: *p* = 1.64 × 10^−3^) and macrophage M1 (Wilcoxon signed-rank test: *p* = 1.7 × 10^−9^) as compared to BLIS tumors. The results indicated that an increase in the ratio of infiltrating anti-tumorigenic to pro-tumorigenic cells is associated with improved MFS. The different survival outcomes and patterns of immune signatures within the four subtypes suggest the necessity to differentiate the extra two intermediate subtypes. Interestingly, although Tfh is considered as anti-tumorigenic and is known to activate B cells to facilitate the anti-tumor response [53], we observed a significantly high abundance of Tfhs in BLIS tumors but low abundance in IM tumors. Considering that the association of various tumor-infiltrating lymphocytes (TILs) with the clinical outcome of cancer has always been controversial [54], distinct immunological mechanisms might take place in different cancer subtypes and under specific conditions. Thus, more studies concerning the origin and development dynamics of TILs on the TNBC subtype level are still needed to improve our understanding of the heterogeneity of TNBC.

## 4. Conclusions and Discussion

Subtyping of TNBC is essential for patient stratification and personalized treatment. Although several studies have investigated this question, their results lack consistency due to the choice of different analytical workflows, sampling methods, and computational tools. By reviewing literature and analyzing public datasets, we reassessed the usage of available tools and presented a guideline for performing molecular subtyping of TNBC (Figure 1), which will be helpful for the rational design of future research. Based on our analysis, we also provided a series of suggestions regarding some issues on either general cancer subtyping or specific to TNBC subtyping. In particular, researchers must pay attention to specific procedures, including the utilization of FFPE samples, batch effect removal across microarray platforms, parameter selection of commonly used computational tools, and the interpretation of Lehmann’s TNBCtype. Note that although most previous TNBC subtyping studies were based on microarrays, RNA sequencing (RNA-seq) technologies have emerged in recent years. Since both RNA-seq and microarray are gene expression profiling methods, their gene expression matrices do not have major differences. As long as researchers follow standard data preprocessing pipelines for RNA-seq data, our results and conclusions should also be applicable to future TNBC subtyping studies based on RNA sequencing (RNA-seq) data.

Furthermore, by properly selecting and integrating public microarray datasets, we performed TNBC subtyping on a larger sample size. Our results validated the four previously discovered subtypes: LAR, MES, IM, and BLIS. Interestingly, these four subtypes overlapped with Burstein’s subtyping results, despite the clustering methods (*k*-means and NMF). Considering the intensive computation of NMF, future studies could use *k*-means to reproduce previous results.

In addition to the four stable and typical subtypes, we report the first discovery of another two subtypes that potentially represent the intermediate states of tumors. Their expression patterns share some similarities with the BLIS and the MES subtypes, but they also differ in some important features, including regulation of immune-related pathways, prognosis, and tumor microenvironment phenotypes.

Previous studies commonly discovered intermediate subtypes without in-depth investigation. Parker et al. [18] found four intermediate subtypes additional to the intrinsic subtypes that resemble luminal tumors, but these subtypes were eliminated from downstream analysis. Lehmann et al. [19] grouped the samples with ambiguous subtype assignments as unclassifiable. The existence of intermediate tumors reflects the heterogeneity of tissues. Breast tumors contain various cell types in addition to the carcinoma cells [55]. Each cluster of cells may demonstrate distinct gene expression patterns. Single-cell RNA-seq further showed that different cells from the same TNBC patient could be assigned to different subtypes [56]. When performing bulk sampling, this heterogeneity might be blurred by averaged expression patterns and result in intermediate states.

Due to their mixed molecular features, intermediate tumors bring challenges to the treatment. Therapies targeting the predominant subclones cannot eliminate other subclones, causing poor treatment effect or tumor recurrence [57]. Thus, we consider the intermediate subtype as a promising research question that has clinical implications. For example, new subtyping methods could be developed based on paired single-cell and bulk samples to better understand local and global features of the heterogeneous tumor microenvironment.

Molecular subtyping for TNBC based on gene expression profiles is still immature for clinical application [49]. Several limitations exist in current studies. First, the current gold standard for molecular testing in clinical settings such as IHC is mostly protein-based. However, previous studies on TNBC subtyping are mostly based on transcriptomics. Intrinsic subtyping shows that basal-like tumors (triple-negative at the mRNA level) cannot overlap with TNBC completely [22], indicating differences between transcriptomic subtypes and proteomic subtypes. A few studies have integrated proteomics into TNBC research [58,59]. Future studies could conduct a more comprehensive analysis on the protein level and develop clinical biomarkers.

Second, understanding of cancer immunology revolutionized cancer treatment in recent years by leading to the development of immunotherapy. TILs have been used as clustering features [10,12]. However, the association between some immune cells and clinical outcomes of TNBC remains controversial. It is recommended to conduct large-scale tumor microenvironment analysis of existing TNBC subtypes and to develop computational methods for identifying clinically targetable immune subtypes based on the features of TILs.

Third, gene expression levels fluctuate under different conditions, but traditional transcriptomic or proteomic data are measured at a fixed time point. Further studies could consider this issue by integrating time-series data and adjusting feature weights based on the range of gene expression changes. Researchers could also use some features that are more resistant to gene expression fluctuations such as gene interaction perturbations [28] and rank-based signatures [60].

## Figures and Tables

**Figure 1 cancers-14-02571-f001:**
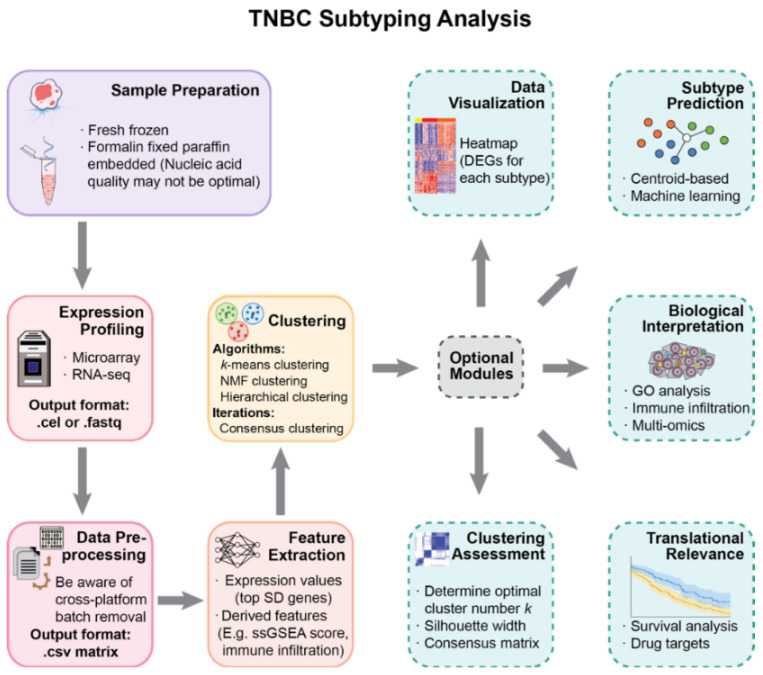
A modularized pipeline for TNBC subtyping. Each step provides alternative methods and data types. Several optional modules are listed for downstream analysis after subtype discovery. This pipeline is also applicable for similar research in other cancer types.

**Figure 2 cancers-14-02571-f002:**
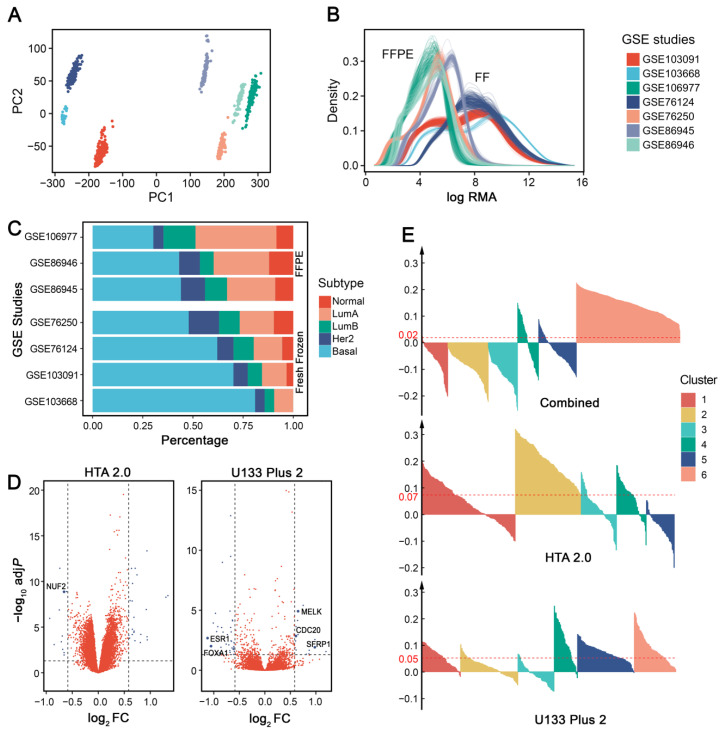
Effects of sample preservation and batch effect removal on TNBC subtyping. (**A**) PCA plot of the 7 datasets (GSE103668, GSE103091, GSE76124, and GSE76250 were generated from FF tissues; GSE86945, GSE86946, and GSE106977 were generated from FFPE samples) without batch effect removal. (**B**) Log-transformed robust multichip average (RMA) value density distribution of each dataset. (**C**) Intrinsic subtyping of 7 datasets using PAM50 centroids. (**D**) Volcano plot of differentially expressed genes between datasets using FFPE and FF tissues. (**E**) Average silhouette width of different batch effect removal schemes. From top to bottom: Combining datasets from HTA 2.0 and U133 Plus 2 by cross-platform batch effect removal; Removing batch effect separately for datasets from HTA 2.0 and U133 Plus 2, respectively. The red lines represent the value of the average silhouette width across all samples.

**Figure 3 cancers-14-02571-f003:**
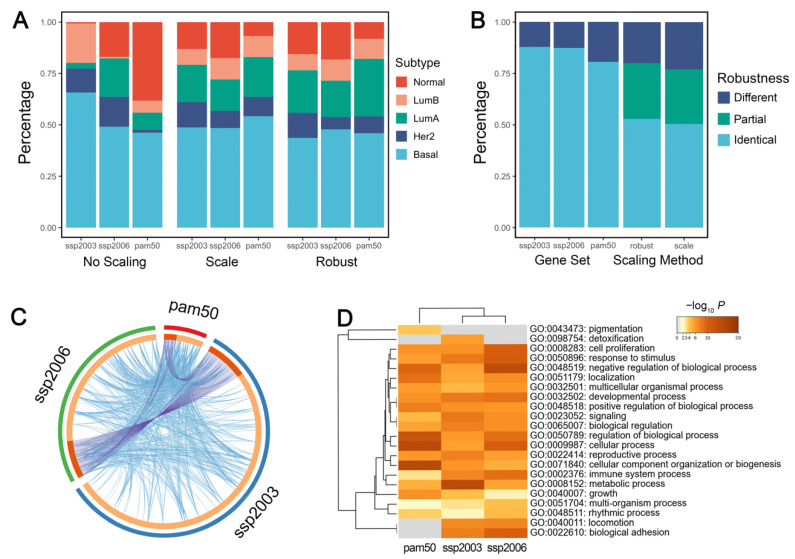
Effects of the centroid gene set and scaling method on intrinsic subtyping. (**A**) Distribution of intrinsic subtypes among gene sets using different scaling methods. (**B**) Robustness of intrinsic subtypes using either different gene sets or scaling methods. When using different parameters, samples with identical intrinsic subtype assignments are labeled ‘identical’. Samples with completely different intrinsic subtype assignments are labeled ‘different’. Sample with only 2 different intrinsic subtype assignments using 3 different gene sets are labeled ‘partial’. (**C**) Circos plot of 3 intrinsic gene sets. The purple line links identical genes and the blue line denotes functional correlations. (**D**) GO pathway enrichment visualized with heatmap. A lower *p* value with a deeper color suggests more genes are enriched in the pathway.

**Figure 4 cancers-14-02571-f004:**
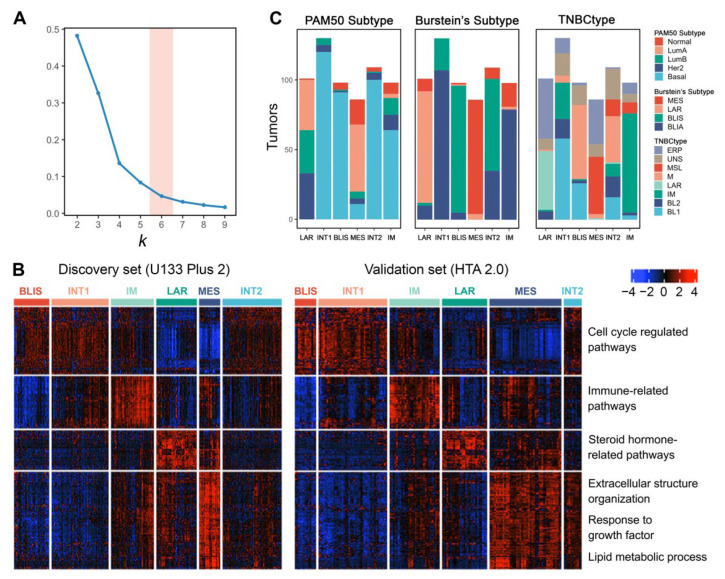
The *k*-means clustering of TNBC with expression profile reveals six stable subtypes. (**A**) The optimal cluster number, 6, is determined as the point where there is a relatively small increase of the area change under the CDF curve with increasing *k*. (**B**) Heatmap shows the distinct gene expression pattern of the 6 named subtypes in both the discovery (457 of 457, U133 Plus 2 platform) and validation sets (165 of 165, HTA 2.0 platform). (**C**) Distribution of other TNBC subtyping results (PAM50 subtypes, Burstein’s subtypes, and Lehmann’s TNBC subtypes) among the 6 subtypes. Samples that could not pass the ER-positive filter of TNBCtype are labeled as ‘ERP’.

**Figure 5 cancers-14-02571-f005:**
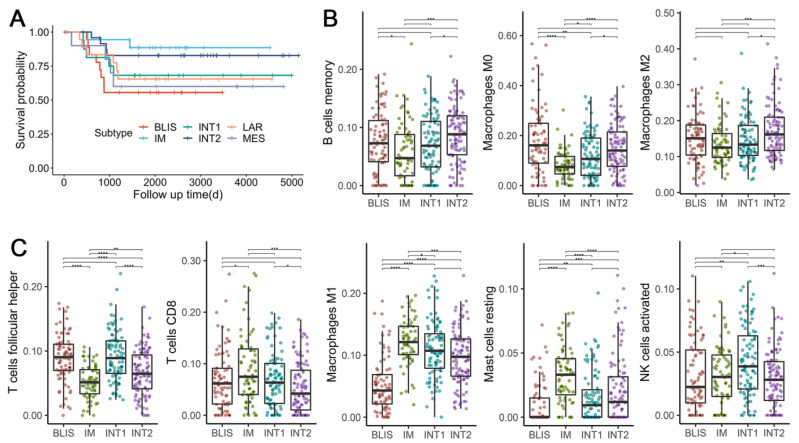
Prognostic differences and immune infiltration patterns of the 6 discovered subtypes. (**A**) The Kaplan–Meier survival curve with MFS from dataset GSE103091. (**B**) Immune infiltration scores of pro-tumorigenic cells among 4 basal-like subtypes. (**C**) Immune infiltration scores of anti-tumorigenic cells among 4 basal-like subtypes. The scores denote the immune cell abundance of the sample. Significance levels are labeled with asterisks (****, *p* < 0.0001; ***, 0.0001 < *p* < 0.001; **, 0.001 < *p* < 0.01; *, 0.01 < *p* < 0.05).

**Table 1 cancers-14-02571-t001:** Clustering algorithms for subtype discovery.

Methods ^a^	Descriptions
**Partitioning-based**	Partitioning clustering iteratively assigns samples between clusters based on their similarity. It is relatively efficient, but sensitive to outliers and needs the number of clusters to be specified in advance.
*k*-means	
*k*-medoids ^b^	
Fuzzy *c*-means [32]	
**Hierarchical clustering**	Hierarchical clustering creates homogeneous groups of samples by either a top-down (divisive) or a bottom-up (agglomerative) approach. The output dendrogram is easy to understand.
Divisive	
Agglomerative	
**Density-based**	Density-based clustering works by detecting densely connected regions. It does not require the number of clusters to be specified and can deal with noisy data and non-convex clusters. However, it is not suitable when there are significant density differences.
DBSCAN ^c^	
OPTICS ^d^ [33]	
DPCA ^e^ [34]	
**Spectral clustering** RatioCut [35]	Spectral clustering is based on graph theory. It reduces the dimensionality of the dataset and then applies a basic clustering algorithm.
Ncut [36]	
**Non-negative matrix factorization** **(NMF)** [30]	NMF reduces the dimension of expression data, and in the meantime, places each sample into a cluster corresponding to the metagene.
**Model-based**	Unlike traditional algorithms, model-based clustering attempts to provide soft assignment and measures the probability of a sample belonging to each cluster.
Gaussian mixture models	
Self-organizing maps [37]	

Note: ^a^ Only lists some representative algorithms are listed here. There are also other clustering algorithm categories and variations to each method. ^b^ Also known as: partition around medoids (PAM). ^c^ DBSCAN: density-based spatial clustering of applications with noise. ^d^ OPTICS: ordering points to identify clustering structure. ^e^ DPCA: density peaks clustering algorithm.

**Table 2 cancers-14-02571-t002:** Algorithms for subtype prediction.

Methods ^a^	Descriptions
**Centroid-based**	Centroid-based methods assign new samples to one of the existing classes based on centroids computed from the discovery set. PAM and ClaNC are examples of modified simple nearest centroid methods.
Simple nearest centroid classifier (SNCC)	
Prediction analysis of microarrays (PAM) [39]	
Classification to nearest centroids (ClaNC) [40]	
**Supervised machine learning**	Supervised machine learning methods can learn rules from labeled training data. For cancer subtyping, machine learning models could be trained on the discovery set and used to predict class assignment on the new datasets. Traditional and simple methods could already achieve good performance.
Support vector machine (SVM)	
*k*-nearest neighbor (*k*-NN)	
Linear discriminant analysis (LDA)	
Decision trees	

Note: ^a^ Only some representative algorithms are listed here. There are also other algorithms and variations for each method.

## Data Availability

All data are publicly available and can be retrieved through the GEO database (Appendix A).

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
