# Peer review of "Reassessment of Reliability and Reproducibility for Triple-Negative Breast Cancer Subtyping"

_cancers, 2022, doi:10.3390/cancers14112571_

Round 1

Reviewer 1 Report

In this manuscript, Yu et al, proposed a schematic pipeline for the TNBC stratification. The authors reexamined the publicly available GEO database for the molecular subtyping of TNBC. They critically analyzed the database and provided several suggestions. The manuscript is very well written, and the data is very well presented. Their suggestions could be helpful for the rational design of future studies.  I have a few minor comments and questions.

  1. Several other IHC confirmed datasets are available publicly that were generated from the TNBC biopsies. What were the criteria for selecting specific GEP Datasets?
  2. Provide more information on GEO datasets, like cancer stage, biopsies sties and mRNA preparation method. This information is important because the authors here suggested paying attention when using FFPE samples so, it is equally important for them to provide the information of the datasets that they used for their study.
  3. Page 3, Line 17: Please clear whether the fold change ≥ 1.5 was after correction of false discovery.
  4. Figure 2B: Labeling GSE with FF and FFPE would be helpful for the readers.
  5. Provide more information on removing batch effects for separate microarray platforms.
  6. What is the gene signature of INT1 and INT2? Since this subtype is mentioned in detail in this study, it would be nice to have a list of a few important genes.

Reviewer 2 Report

I read with great interest the article "Reassessment of reliability and reproducibility for triple-negative breast cancer subtyping” by Xinjian Yu , Yongjing Liu , Siqi Lai and Ming Chen

In my opinion, the article is well-written, structured and the material is well-chosen. Results correctly presented and extensively visualized. Discussion was conducted well.
The article presents the current knowledge on the very important topic of breast cancer.
In terms of research, I rate the work very highly. I believe that the work is fully understandable and does not require any corrections.

I believe that the article is suitable for printing in its current form.

Author Response

Response: Thank you. We sincerely appreciate your high recognition of our work.

Reviewer 3 Report

This work presents recent findings in triple negative breast cancer subtyping - very heterogeneous disease. The problem of molecular classification (subtyping) is solved by integration of available data obtained by different microarray and sequencing platforms.

The paper is well-organized; it is based on extended analysis of microarray datasets using modern bioinformatics tools.

I have only some minor remarks:

Line 15: ‘various prognoses’ - I suggest change wording ‘various’ to ‘poor’ or ‘diverse, often poor prognoses’

Line 33: ‘Keywords’ - may add keyword ‘microarrays’ since significant part of work used this platform, may add ‘pipeline’ since new rather universal computer pipeline was tested in this paper.

Figure 1. Scheme is good, but might update the blocks visualization before ‘Optional modules’ by color or shape (use rectangles or oval, or diamonds). Currently there are two groups of blocks - before ’Optional models’ and after it. May at least mark preliminary steps ‘Sample preparation’, ‘Expression profiling’, ‘Data Preprocessing’, ‘Feature extraction’ and ‘Clustering’ by some color to separate it visually from the optional (additional) modules.

Name of the pipeline ‘TNBC Subtyping analysis’ could be moved up, before the blocks.

Line 73: ‘2.1. Data collection’ - there are number 7 for the datasets, and 4 plus 4 for different dataset types *8 in total). Assume 1 dataset was removed later. Please comment.

Line 76: ‘FF tissues’ - please give abbreviation FF in full.

Line 82: ‘platform HTA 2.0’ - add reference here, give name HTA in full.

Line 94: ‘CDF’ - give the abbreviation in full.

Line 105: ‘PAM50 classifier’ - add reference for PAM50 (it is given in the end of manuscript only)

Line 106: ‘TNBCtype’ - give the abbreviation in full.

Line 108: ‘...they discovered and we downloaded...’ - separate this phrase into 2 sentences

Line 109: ‘We assigned samples’ - add ‘all the samples’, add ‘using this signature’. The phrase is not clear in current form. How the Pearson correlation was measured?

Line 112: ‘MFS records’ - please give abbreviation MFS in full.

Line 205: Table 1. Right column has extra empty cells (horizontal lines). Please try to reformat or fix it.

Thus for Spectral clustering row - there is no additional horizontal line at right.

Line 206: may add wording ‘Note’ after the table. Keep all the table notes on same page, in smaller font; separate by empty line from the next text. It is not convenient to read, mixed with main text now.

Same remark is for Table 2. (Line 229). There are several table comments (notes) - a-f, in fact for each raw. Might include these comments (full method names) into the table.

Formally the reference [34] in the table is given before the reference [33] in the main text. Please fix the order, or mark these references again in the text, not only in the table.

Line 351: ‘.’. - extra signs.

Line 493: ‘is still premature for clinical application.’ - need a reference her. Or write that it is authors’ statement based on this paper.

Line 529: ‘GEO’ - it is good give exact references to the datasets here, not just common GEO name.

Reviewer 4 Report

Yu and colleagues present yet another look on the subject of TNBC classification based on publicly available datasets. They demonstrate the problems and limitations with previous work in that field and identify two new states of TNBC tumors in addition to the previously described four subtypes. Although this manuscript has potential merit for the community it seems premature in its conclusions and lacks substantial novelty. Therefore I cannot recommend publication in Cancers.

Author Response

Response: Thanks for this comment. TNBC has long been a hot topic in the field of cancer research. In this context, it is difficult to conduct a completely novel study. Nevertheless, we have summarized several findings that may help other researchers, while other reviewers have also recognized the significance of this work. We believe this work is of particular importance for researchers.

Furthermore, the molecular subtyping of TNBC has been controversial, which greatly limits the development of personalized therapies. The lack of consensus is likely due to limited sample size for each individual research group, and great discrepancies in technical and computational methods. We provided new insights on this issue by large-scale computational analysis with comprehensive assessment of current methods and integration of datasets. We believe it helps to standardize TNBC subtyping pipelines and improve the reproducibility of TNBC subtypes.